# Phase-aware Speech Enhancement with Deep Complex U-Net

**Hyeong-Seok Choi**[1,3], **Jang-Hyun Kim**[2,3], **Jaesung Huh**[3], **Adrian Kim**[3],
**Jung-Woo Ha**[3], **Kyogu Lee**[1]
[1]Department of Transdisciplinary Studies, Seoul National University, Seoul, Korea
[2]Department of Mathematical Sciences, Seoul National University, Seoul, Korea
[3]Clova AI Research, NAVER Corp., Seongnam, Korea
kekepa15@snu.ac.kr, blue378@snu.ac.kr, jaesung.huh@navercorp.com,
adrian.kim@navercorp.com, jungwoo.ha@navercorp.com, kglee@snu.ac.kr

## Abstract

Most deep learning-based models for speech enhancement have mainly focused on estimating the magnitude of spectrogram while reusing the phase from noisy speech for reconstruction. This is due to the difficulty of estimating the phase of clean speech. To improve speech enhancement performance, we tackle the phase estimation problem in three ways. First, we propose Deep Complex U-Net, an advanced U-Net structured model incorporating well-defined complex-valued building blocks to deal with complex-valued spectrograms. Second, we propose a polar coordinate-wise complex-valued masking method to reflect the distribution of complex ideal ratio masks. Third, we define a novel loss function, weighted source-to-distortion ratio (wSDR) loss, which is designed to directly correlate with a quantitative evaluation measure. Our model was evaluated on a mixture of the Voice Bank corpus and DEMAND database, which has been widely used by many deep learning models for speech enhancement. Ablation experiments were conducted on the mixed dataset showing that all three proposed approaches are empirically valid. Experimental results show that the proposed method achieves state-of-the-art performance in all metrics, outperforming previous approaches by a large margin[1].

## 1 Introduction

Speech enhancement is one of the most important and challenging tasks in speech applications where the goal is to separate clean speech from noise when noisy speech is given as an input. As a fundamental component for speech-related systems, the applications of speech enhancement vary from speech recognition front-end modules to hearing aid systems for the hearing-impaired (Erdogan et al., 2015; Weninger et al., 2015; Wang, 2017).

Due to recent advances in deep learning, the speech enhancement task has been able to reach high levels in performance through significant improvements. When using audio signals with deep learning models, it has been a common practice to transform a time-domain waveform to a time-frequency (TF) representation (i.e. spectrograms) via short-time-Fourier-transform (STFT). Spectrograms are represented as complex matrices, which are normally decomposed into magnitude and phase components to be used in real-valued networks. In tasks involving audio signal reconstruction, such as speech enhancement, it is ideal to perform correct estimation of both components. Unfortunately, complex-valued phase has been often neglected due to the difficulty of its estimation. This has led to the situation where most approaches focus only on the estimation of a magnitude spectrogram while reusing noisy phase information (Huang et al., 2014; Xu et al., 2015; Grais et al., 2016; Nugraha et al., 2016; Takahashi et al., 2018b). However, reusing phase from noisy speech has clear limitations, particularly under extremely noisy conditions, in other words, when signal-to-noise ratio (SNR) is low. This can be easily verified by simply using the magnitude spectrogram of clean

---

[1]Audio samples are available in the following link: http://kkp15.github.io/DeepComplexUnet

speech with the phase spectrogram of noisy speech to reconstruct clean speech, as illustrated in Figure 1. We can clearly see that the difference between clean and estimated speech signals gets larger as the input SNR gets lower.

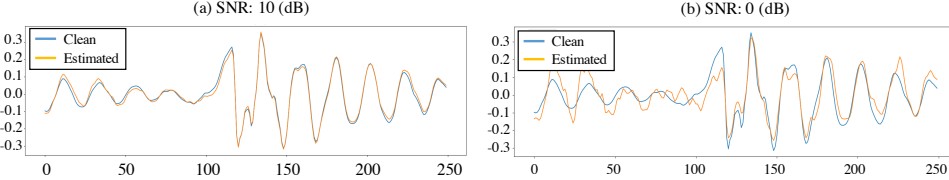

Figure 1: Effects of reusing phase of noisy speech under different SNR conditions.

A popular approach to speech enhancement is to optimize a mask which produces a spectrogram of clean speech when applied to noisy input audio. One of the first mask-based attempts to perform the task by incorporating phase information was the proposal of the phase-sensitive mask (PSM) (Erdogan et al., 2015). Since the performance of PSM was limited because of reusing noisy phase, later studies proposed using complex-valued ratio mask (cRM) to directly optimize on complex values (Williamson et al., 2016; Ephrat et al., 2018). We found this direction promising for phase estimation because it has been shown that a complex ideal ratio mask (cIRM) is guaranteed to give the best oracle performance out of other ideal masks such as ideal binary masks, ideal ratio masks, or PSMs (Wang et al., 2016). Moreover, this approach jointly estimates magnitude and phase, removing the need of separate models. To estimate a complex-valued mask, a natural desire would be to use an architecture which can handle complex-domain operations. Recent work gives a solution to this by providing deep learning building blocks adapted to complex arithmetic (Trabelsi et al., 2018).

In this paper, we build upon previous studies to design a new complex-valued masking framework, based on a proposed variant of U-Net (Ronneberger et al., 2015), named Deep Complex U-Net (**DCUnet**). In our proposed framework, DCUnet is trained to estimate a complex ratio mask represented in polar coordinates with prior knowledge observable from ideal complex-valued masks. With the complex-valued estimation of clean speech, we can use inverse short-time-Fourier-transform (ISTFT) to convert a spectrogram into a time-domain waveform. Taking this as an advantage, we introduce a novel loss function which directly optimizes source-to-distortion ratio (SDR) (Vincent et al., 2006), a quantitative evaluation measure widely used in many source separation tasks.

Our contributions can be summarized as follows:

1. We propose a new neural architecture, **Deep Complex U-Net**, which combines the advantages of both deep complex networks and U-Net, yielding state-of-the-art performance.
2. While pointing out limitations of current masking strategies, we design a new complex-valued masking method based on polar coordinates.
3. We propose a new loss function **weighted-SDR loss**, which directly optimizes a well known quantitative evaluation measure.

## 2 RELATED WORKS

Phase estimation for audio signal reconstruction has been a recent major interest within the audio source separation community because of its importance and difficulty. While iterative methods such as the Griffin-Lim algorithm and its variants (Griffin & Lim, 1984; Perraudin et al., 2013) aimed to address this problem, neural network-based approaches are recently attracting attention as non-iterative alternatives.

One major approach is to use an end-to-end model that takes audio as raw waveform inputs without using any explicit time-frequency (TF) representation computed via STFT (Pascual et al., 2017; Rethage et al., 2018; Stoller et al., 2018; Germain et al., 2018). Since raw waveforms inherently contain phase information, it is expected to achieve phase estimation naturally. Another method is to estimate magnitude and phase using two separate neural network modules which serially estimate magnitude and phase (Afouras et al., 2018; Takahashi et al., 2018a). In this framework, the phase

estimation module uses noisy phase with predicted magnitude to estimate phase of clean speech. There is also a recent study which proposed to use additional layers with trainable discrete values for phase estimation (Roux et al., 2018).

A more straightforward method would be to jointly estimate magnitude and phase by using a continuous complex-valued ratio mask (cRM). Previous studies tried this joint estimation approach bounding the range of the cRM (Williamson et al., 2016; Ephrat et al., 2018). Despite the advantages of the cRM approach, previously proposed methods had limitations with regard to the loss function and the range of the mask which we will be returning with more details in Section 3 along with our proposed methods to alleviate these issues. As a natural extension to the works above, some studies have also undergone to examine whether complex-valued networks are useful when dealing with intrinsically complex-valued data. In the series of two works, complex-valued networks were shown to help singing voice separation performance with both fully connected neural networks and recurrent neural networks (Lee et al., 2017a;b). However, the approaches were limited as it ended up only switching the real-valued network into a complex-valued counterpart and leaving the other deep learning building blocks such as weight initialization and normalization technique in a real-valued manner. Also, the works do not show whether the phase was actually well estimated either quantitatively or qualitatively, only ending up showing that there was a performance gain.

## 3 PHASE-AWARE SPEECH ENHANCEMENT

In this section we will provide details on our approach, starting with our proposed model Deep Complex U-Net, followed by the masking framework based on the model. Finally, we will introduce a new loss function to optimize our model, which takes a critical role for proper phase estimation.

Before getting into details, here are some notations used throughout the paper. The input mixture signal $x(n) = y(n) + z(n) \in \mathbb{R}$ is assumed to be a linear sum of the clean speech signal $y(n) \in \mathbb{R}$ and noise $z(n) \in \mathbb{R}$, where estimated speech is denoted as $\hat{y}(n) \in \mathbb{R}$. Each of the corresponding time-frequency $(t, f)$ representations computed by STFT is denoted as $X_{t,f} \in \mathbb{C}$, $Y_{t,f} \in \mathbb{C}$, $Z_{t,f} \in \mathbb{C}$, and $\hat{Y}_{t,f} \in \mathbb{C}$. The ground truth mask cIRM is denoted as $M_{t,f} \in \mathbb{C}$ and the estimated cRM is denoted as $\hat{M}_{t,f} \in \mathbb{C}$, where $M_{t,f} = Y_{t,f}/X_{t,f}$.

### 3.1 DEEP COMPLEX U-NET

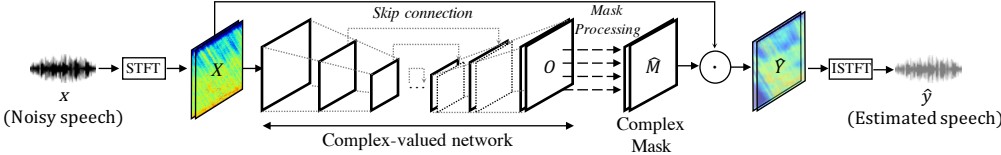

Figure 2: Illustration of speech enhancement framework with DCUnet.

The U-Net structure is a well known architecture composed as a convolutional autoencoder with skip-connections, originally proposed for medical imaging in computer vision community (Ronneberger et al., 2015). Furthermore, the use of real-valued U-Net has been shown to be also effective in many recent audio source separation tasks such as music source separation (Jansson et al., 2017; Stoller et al., 2018; Takahashi et al., 2018b), and speech enhancement (Pascual et al., 2017). Deep Complex U-Net (DCUnet) is an extended U-Net, refined specifically to explicitly handle complex domain operations. In this section, we will describe how U-Net is modified using the complex building blocks originally proposed by (Trabelsi et al., 2018).

**Complex-valued Building Blocks.** Given a complex-valued convolutional filter $W = A + iB$ with real-valued matrices $A$ and $B$, the complex convolution operation on complex vector $h = x + iy$ with $W$ is done by $W * h = (A * x - B * y) + i(B * x + A * y)$. In practice, complex convolutions can be implemented as two different real-valued convolution operations with shared real-valued convolution filters. Details are illustrated in Appendix A.

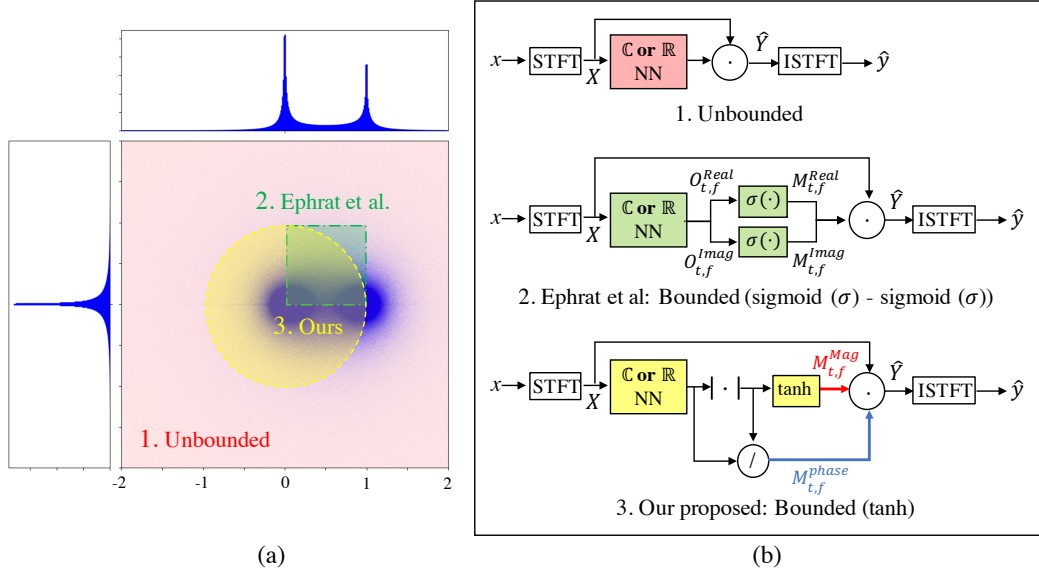

Figure 3: Illustration of three cRM methods and their corresponding output range in complex space. The Neural Network (NN) can be either DCUnet ($\mathbb{C}$) or a corresponding real-valued U-Net ($\mathbb{R}$) with the same parameter size. Three different output ranges of cRM methods (1, 2 and 3) in (b) are shown in (1, 2 and 3) in (a). 1. Unbounded mask: a masking method without bounded outputs. 2. Bounded (sigmoid-sigmoid) mask: The proposed method by Ephrat et al. 3. Bounded (tanh) mask: Our proposed masking method. The detailed model specification is described in Appendix B.

Activation functions like ReLU were also adapted to the complex domain. In previous work, $\mathbb{C}$ReLU, an activation function which applies ReLU on both real and imaginary values, was shown to produce the best results out of many suggestions. Details on batch normalization and weight initialization for complex networks can be found in Trabelsi et al. (2018).

**Modifying U-Net.** The proposed Deep Complex U-Net is a refined U-Net architecture applied in STFT-domain. Modifications done to the original U-Net are as follows. Convolutional layers of U-Net are all replaced to complex convolutional layers, initialized to meet the Glorot's criteria (Glorot & Bengio, 2010). Here, the convolution kernels are set to be independent to each other by initializing the weight tensors as unitary matrices for better generalization and fast learning (Cogswell et al., 2015). Complex batch normalization is implemented on every convolutional layer except the last layer of the network. In the encoding stage, max pooling operations are replaced with strided complex convolutional layers to prevent spatial information loss. In the decoding stage, strided complex deconvolutional operations are used to restore the size of input. For the activation function, we modified the previously suggested $\mathbb{C}$ReLU into leaky $\mathbb{C}$ReLU, where we simply replace ReLU into leaky ReLU (Maas et al., 2013), making training more stable. Note that all experiments performed in Section 4 are done with these modifications.

## 3.2 COMPLEX-VALUED MASKING ON POLAR COORDINATES

As our proposed model can handle complex values, we aim to estimate cRM for speech enhancement. Although it is possible to directly estimate the spectrogram of a clean source signal, it has been shown that better performance can be achieved by applying a weighting mask to the mixture spectrogram (Wang et al., 2014). One thing to note is that real-valued ratio masks (RM) only change the scale of the magnitude without changing phase, resulting in irreducible errors as illustrated in Appendix D. On the other hand, cRM also perform a rotation on the polar coordinates, allowing to correct phase errors. In other words, the estimated speech spectrogram $\hat{Y}_{t,f}$ is computed by multiplying the estimated mask $\hat{M}_{t,f}$ on the input spectrogram $X_{t,f}$ as follows:

$$\hat{Y}_{t,f} = \hat{M}_{t,f} \cdot X_{t,f} = \left| \hat{M}_{t,f} \right| \cdot \left| X_{t,f} \right| \cdot e^{i(\theta_{\hat{M}_{t,f}} + \theta_{X_{t,f}})} \tag{1}$$

In this state, the real and imaginary values of the estimated cRM is unbounded. Although estimating an unbounded mask makes the problem well-posed (see Appendix D for more information), we can imagine the difficulty of optimizing from an infinite search space compared to a bounded one.

Therefore, a few techniques have been tried to bound the range of cRM. For example, Williamson et al. tried to directly optimize a complex mask into a cIRM compressed to a heuristic bound (Williamson et al., 2016). However, this method was limited since it was only able to succeed in training the model by computing the error between cIRM and the predicted cRM which often leads to a degradation of performance (Wang et al., 2014; Yu et al., 2017). More recently, Ephrat et al. proposed a *rectangular coordinate-wise* cRM made with sigmoid compressions onto each of the real and imaginary parts of the output of the model (Ephrat et al., 2018). After then MSE between clean source $Y$ and estimated source $\hat{Y}$ was computed in STFT-domain to train the model. However, the proposed masking method has two main problems regarding phase estimation. First, it suffers from the inherent problem of not being able to reflect the distribution of cIRM as shown in Figure 3 and Appendix E. Second, this approach results in a cRM with a restricted rotation range of $0°$ to $90°$ (only clock-wise), which makes it hard to correct noisy phase.

To alleviate these problems, we propose a *polar coordinate-wise* cRM method that imposes non-linearity only on the magnitude part. More specifically, we use a hyperbolic tangent non-linearity to bound the range of magnitude part of the cRM be $[0, 1)$ which makes the mask bounded in an unit-circle in complex space. The corresponding phase mask is naturally obtained by dividing the output of the model with the magnitude of it. More formally, let $g(\cdot)$ be our neural network and the output of it be $O_{t,f} = g(X_{t,f})$. The proposed complex-valued mask $\hat{M}_{t,f}$ is estimated as follows:

$$\hat{M}_{t,f} = \left| \hat{M}_{t,f} \right| \cdot e^{i\theta_{\hat{M}_{t,f}}} = \hat{M}_{t,f}^{mag} \cdot \hat{M}_{t,f}^{phase} \tag{2}$$

$$\hat{M}_{t,f}^{mag} = \begin{cases} tanh(|O_{t,f}|) & \text{(bounded cond.)} \\ |O_{t,f}| & \text{(unbounded cond.)} \end{cases}, \quad \hat{M}_{t,f}^{phase} = O_{t,f}/|O_{t,f}| \tag{3}$$

A summarized illustration of cRM methods is depicted in Figure 3.

### 3.3 WEIGHTED-SDR LOSS

A popular loss function for audio source separation is mean squared error (MSE) between clean source $Y$ and estimated source $\hat{Y}$ on the STFT-domain. However, it has been reported that optimizing the model with MSE in complex STFT-domain fails in phase estimation due to the randomness in phase structure (Williamson et al., 2016). As an alternative, it is possible to use a loss function defined in the time-domain instead, as raw waveforms contain inherent phase information. While MSE on waveforms can be an easy solution, we can expect it to be more effective if the loss function is directly correlated with well-known evaluation measures defined in the time-domain.

Here, we propose an improved loss function **weighted-SDR loss** by building upon a previous work which attempts to optimize a standard quality measure, source-to-distortion ratio (SDR) (Venkataramani et al., 2017). The original loss function $loss_{Ven}$ suggested by Venkataramani et al. is formulated upon the observation from Equation 4, where $y$ is the clean source signal and $\hat{y}$ is the estimated source signal. In practice, the negative reciprocal is optimized as in Equation 5.

$$\max_{\hat{y}} \ SDR(y, \hat{y}) := \max_{\hat{y}} \frac{<y, \hat{y}>^2}{||y||^2 ||\hat{y}||^2 - <y, \hat{y}>^2} \propto \min_{\hat{y}} \frac{||\hat{y}||^2}{<y, \hat{y}>^2} \tag{4}$$

$$loss_{Ven}(y, \hat{y}) := -\frac{<y, \hat{y}>^2}{||\hat{y}||^2} \tag{5}$$

Although using Equation 5 works as a loss function, there are a few critical flaws in the design. First, the lower bound becomes $-||y||^2$, which depends on the value of $y$ causing fluctuation in the

loss values when training. Second, when the target $y$ is empty (i.e., $y = 0$) the loss becomes zero, preventing the model to learn from noisy-only data due to zero gradients. Finally, the loss function is not scale sensitive, meaning that the loss value is the same for $\hat{y}$ and $c\hat{y}$, where $c \in \mathbb{R}$.

To resolve these issues, we redesigned the loss function by giving several modifications to Equation 5. First, we made the lower bound of the loss function independent to the source $y$ by restoring back the term $\|y\|^2$ and applying square root as in Equation 6. This makes the loss function bounded within the range [-1, 1] and also be more phase sensitive, as inverted phase gets penalized as well.

$$loss_{SDR}(y, \hat{y}) := -\sqrt{-\frac{loss_{Ven}}{\|y\|^2}} = -\frac{<y, \hat{y}>}{\|y\|\|\hat{y}\|} \tag{6}$$

Expecting to be complementary to source prediction and to propagate errors for noise-only samples, we also added a noise prediction term $loss_{SDR}(z, \hat{z})$. To properly balance the contributions of each loss term and solve the scale insensitivity problem, we weighted each term proportional to the energy of each signal. The final form of the suggested weighted-SDR loss is as follows:

$$loss_{wSDR}(x, y, \hat{y}) := \alpha\, loss_{SDR}(y, \hat{y}) + (1 - \alpha)loss_{SDR}(z, \hat{z}) \tag{7}$$

where, $\hat{z} = x - \hat{y}$ is estimated noise and $\alpha = \|y\|^2/(\|y\|^2 + \|z\|^2)$ is the energy ratio between clean speech $y$ and noise $z$. Note that although weighted SDR loss is a time-domain loss function, it can be backpropagated through our framework. Specifically, STFT and ISTFT operations are implemented as 1-D convolution and deconvolution layers consisting of fixed filters initialized with the discrete Fourier transform matrix. The detailed properties of the proposed loss function are in Appendix C.

## 4 EXPERIMENTS

### 4.1 EXPERIMENT SETUP

**Dataset.** For all experiments, we used the same experimental setups as previous works in order to perform direct performance comparison (Pascual et al., 2017; Rethage et al., 2018; Soni et al., 2018; Germain et al., 2018). Noise and clean speech recordings were provided from the Diverse Environments Multichannel Acoustic Noise Database (DEMAND) (Thiemann et al., 2013) and the Voice Bank corpus (Veaux et al., 2013), respectively, each recorded with sampling rate of 48kHz. Mixed audio inputs used for training were composed by mixing the two datasets with four signal-to-noise ratio (SNR) settings (15, 10, 5, and 0 (dB)), using 10 types of noise (2 synthetic + 8 from DEMAND) and 28 speakers from the Voice Bank corpus, creating 40 conditional patterns for each speech sample. The test set inputs were made with four SNR settings different from the training set (17.5, 12.5, 7.5, and 2.5 (dB)), using the remaining 5 noise types from DEMAND and 2 speakers from the Voice Bank corpus. Note that the speaker and noise classes were uniquely selected for the training and test sets.

**Pre-processing.** The original raw waveforms were first downsampled from 48kHz to 16kHz. For the actual model input, complex-valued spectrograms were obtained from the downsampled waveforms via STFT with a 64ms sized Hann window and 16ms hop length.

**Implementation.** All experiments were implemented and fine-tuned with NAVER Smart Machine Learning (NSML) platform (Sung et al., 2017; Kim et al., 2018).

### 4.2 COMPARISON RESULTS

In this subsection, we compare overall speech enhancement performance of our method with previously proposed algorithms. As a baseline approach, Wiener filtering (**Wiener**) with a priori noise SNR estimation was used, along with recent deep-learning based models which are briefly described as the following: **SEGAN**: a *time-domain* U-Net model optimized with generative adversarial networks. **Wavenet**: a *time-domain* non-causal dilated wavenet-based network. **MMSE-GAN**: a *time-frequency* masking-based method with modified adversarial training method. **Deep Feature Loss**: a *time-domain* dilated convolution network trained with feature loss from a classifier network.

---

[2]https://www.crcpress.com/downloads/K14513/K14513_CD_Files.zip

Table 1: Quantitative evaluation results of other algorithms and proposed methods (DCUnet-20, Large-DCUnet-20). Higher score means better performance where bold text indicates highest score per evaluation measure. **CSIG**: Mean opinion score (MOS) predictor of signal distortion **CBAK**: MOS predictor of background-noise intrusiveness **COVL**: MOS predictor of overall signal quality **PESQ**: Perceptual evaluation of speech quality **SSNR**: Segmental SNR. All evaluation measures were computed by using open source implementation.[2]

|  | CSIG | CBAK | COVL | PESQ | SSNR |
|---|---|---|---|---|---|
| Wiener (Scalart et al., 1996) | 3.23 | 2.68 | 2.67 | 2.22 | 5.07 |
| SEGAN (Pascual et al., 2017) | 3.48 | 2.94 | 2.80 | 2.16 | 7.73 |
| Wavenet (Rethage et al., 2018) | 3.62 | 3.23 | 2.98 | - | - |
| MMSE-GAN (Soni et al., 2018) | 3.80 | 3.12 | 3.14 | 2.53 | - |
| Deep Feature Loss (Germain et al., 2018) | 3.86 | 3.33 | 3.22 | - | - |
| DCUnet-20 (ours) | 4.24 | 4.00 | 3.69 | 3.13 | 15.95 |
| Large-DCUnet-20 (ours) | **4.34** | **4.10** | **3.81** | **3.24** | **16.85** |

For comparison, we used the configuration of using a 20-layer Deep Complex U-Net (**DCUnet-20**) to estimate a tanh bounded cRM, optimized with weighted-SDR loss. As a showcase for the potential of our approach, we also show results from a larger DCUnet-20 (**Large-DCUnet-20**) which has more channels in each layer. Both architectures are specified in detail in Appendix B. Results show that our proposed method outperforms the previous state-of-the-art methods with respect to all metrics by a large margin. Additionally, we can also see that larger models yield better performance. We see the reason to this significant improvement coming from the phase estimation quality of our method, which we plan to investigate in later sections.

## 4.3 ABLATION STUDIES

Table 2: Table of quantitative evaluation results with corresponding mask and loss function in three different model configurations (DCU-10, DCU-16 and DCU-20). The bold font indicates the best loss function when fixing the masking method. The underline indicates the best masking method when fixing the loss function.

|  | CSIG | | | CBAK | | | COVL | | | PESQ | | | SSNR | | |
|---|---|---|---|---|---|---|---|---|---|---|---|---|---|---|---|
| **DCU-10** | Spc | Wav | wSDR | Spc | Wav | wSDR | Spc | Wav | wSDR | Spc | Wav | wSDR | Spc | Wav | wSDR |
| UBD | 3.51 | 3.71 | **3.72** | 3.50 | 3.52 | **3.56** | 3.13 | 3.15 | **3.19** | **2.78** | 2.61 | 2.67 | 11.86 | 13.18 | **13.37** |
| BDSS | 2.94 | 3.19 | **3.21** | **3.43** | 3.35 | 3.39 | 2.81 | 2.85 | **2.89** | **2.79** | 2.61 | 2.67 | 11.61 | 12.30 | **12.30** |
| BDT | 3.30 | 3.74 | **3.72** | 3.52 | 3.57 | **3.60** | 3.02 | 3.19 | **3.22** | **2.79** | 2.66 | 2.72 | 12.30 | 13.54 | **13.60** |
| **DCU-16** | Spc | Wav | wSDR | Spc | Wav | wSDR | Spc | Wav | wSDR | Spc | Wav | wSDR | Spc | Wav | wSDR |
| UBD | 3.97 | 4.00 | **4.07** | 3.66 | 3.72 | **3.75** | 3.46 | 3.43 | **3.48** | **2.96** | 2.85 | 2.93 | 12.75 | 14.25 | **14.44** |
| BDSS | 3.63 | 3.57 | **3.72** | 3.59 | 3.51 | **3.56** | **3.28** | 3.15 | 3.27 | **2.98** | 2.82 | 2.87 | 12.05 | 12.89 | **12.89** |
| BDT | 3.97 | 4.03 | **4.10** | 3.70 | 3.74 | **3.77** | 3.49 | 3.45 | **3.52** | **3.01** | 2.88 | 2.93 | 12.96 | 14.39 | **14.44** |
| **DCU-20** | Spc | Wav | wSDR | Spc | Wav | wSDR | Spc | Wav | wSDR | Spc | Wav | wSDR | Spc | Wav | wSDR |
| UBD | 4.02 | 4.15 | **4.25** | 3.88 | 3.93 | **4.01** | 3.57 | 3.59 | **3.70** | 3.11 | 3.01 | **3.13** | 14.81 | 16.03 | **16.36** |
| BDSS | 3.77 | 3.74 | **3.93** | 3.68 | 3.65 | **3.72** | 3.40 | 3.33 | **3.47** | 3.06 | 2.98 | **3.05** | 12.80 | 13.66 | **13.68** |
| BDT | 4.02 | 4.18 | **4.24** | 3.87 | 3.95 | **4.00** | 3.58 | 3.64 | **3.69** | **3.13** | 3.08 | **3.13** | 14.59 | 15.85 | **15.95** |

**Masking strategy and loss functions.** In this experiment, we show the evaluation results on how the various masking strategies and loss functions affect the performance of speech enhancement. Three masking strategies (Unbounded (UBD); Ephrat et al.: Bounded (sig-sig) (BDSS); Bounded (tanh) (BDT)) are compared to see the effect of each different masking method. Then, to compare

our proposed loss function (weighted-SDR) with MSE, we compared two MSE in STFT-domain and time-domain (Spectrogram-MSE (Spc); Wave-MSE (Wav); and proposed: weighted-SDR (wSDR)).

Table 2 shows the jointly combined results on varied masking strategies and loss functions, where three models (DCU-10 (1.4M), DCU-16 (2.3M), and DCU-20 (3.5M)) are investigated to see how architectural differences in the model affect quantitative results. In terms of masking strategy, the proposed BDT mask mostly yields better results than UBD mask in DCU-10 and DCU-16, imply- ing the importance of limiting optimization space with prior knowledge. However, in the case of DCU-20, UBD mask was able to frequently surpass the performance of BDT mask. Intuitively, this indicates that when the number of parameter gets large enough, the model is able to fit the distri- bution of data well even when the optimization space is not bounded. In terms of the loss function, almost every result shows that optimizing with wSDR loss gives the best result. However, we found out that Spc loss often provides better PESQ results than wSDR loss for DCU-10 and DCU-16 except DCU-20 case where Spc and wSDR gave similar PESQ results.

Table 3: Table of quantitative evaluation results from three different settings (cRM$\mathbb{C}$n: Complex- valued output/Complex-valued network, cRM$\mathbb{R}$n: Complex-valued output/Real-valued network, and RM$\mathbb{R}$n: Real-valued output/Real-valued network) to show the appropriateness of using complex- valued networks for speech enhancement. Bold font indicates the best results.

| | 20-layer | | | 16-layer | | | 10-layer | | |
|---|---|---|---|---|---|---|---|---|---|
| | cRM$\mathbb{C}$n | cRM$\mathbb{R}$n | RM$\mathbb{R}$n | cRM$\mathbb{C}$n | cRM$\mathbb{R}$n | RM$\mathbb{R}$n | cRM$\mathbb{C}$n | cRM$\mathbb{R}$n | RM$\mathbb{R}$n |
| **CSIG** | **4.24** | 4.21 | 4.06 | **4.10** | 4.06 | 3.88 | **3.74** | 3.71 | 3.71 |
| **CBAK** | **4.00** | 3.92 | 3.40 | **3.77** | 3.75 | 3.33 | **3.60** | 3.59 | 3.23 |
| **COVL** | **3.69** | 3.65 | 3.40 | **3.52** | 3.49 | 3.26 | **3.22** | 3.20 | 3.01 |
| **PESQ** | **3.13** | 3.07 | 2.74 | **2.93** | 2.91 | 2.66 | **2.72** | **2.72** | 2.51 |
| **SSNR** | **15.95** | 15.54 | 9.90 | **14.44** | 14.25 | 9.66 | **13.60** | 13.49 | 9.43 |

**Validation on complex-valued network and mask.** In order to show that complex neural networks are effective, we compare evaluation results of DCUnet ($\mathbb{C}$n) and its corresponding real-valued U- Net setting with the same parameter size ($\mathbb{R}$n). For the real-valued network, we tested two settings cRM$\mathbb{R}$n and RM$\mathbb{R}$n to show the effectiveness of phase estimation. The first setting takes a complex- valued spectrogram as an input, estimating a complex ratio mask (cRM) with a tanh bound. The second setting takes a magnitude spectrogram as an input, estimating a magnitude ratio mask (RM) with a sigmoid bound. All models were trained with weighted-SDR loss, where the ground truth phase was given while training RM$\mathbb{R}$n. Additionally, all models were trained on different number of parameters (20-layer (3.5M), 16-layer (2.3M), and 10-layer (1.4M)) to show that the results are consistent regardless of model capacity. Detailed network architectures for each model are illustrated in Appendix B.

In Table 3, evaluation results show that our approach cRM$\mathbb{C}$n makes better results than conventional method RM$\mathbb{R}$n for all cases, showing the effectiveness of phase correction. Also, cRM$\mathbb{C}$n gives bet- ter results than cRM$\mathbb{R}$n, which indicates that using complex-valued networks consistently improve the performance of the network. Note that these results are consistent through every evaluation mea- sure and model size.

### 4.4 QUALITATIVE EVALUATION

We performed qualitative evaluations by obtaining preference scores between the proposed DCUnet (Large-DCUnet-20) and baseline methods. 15 utterance samples with different noise levels were se- lected from the test set and used for subjective listening tests. For each noisy sample, all possible six pairs of denoised audio samples from four different algorithms were presented to the participants in a random order, resulting in 90 pairwise comparisons to be made by each subject. For each compar- ison, participants were presented with three audio samples - original noisy speech and two denoised speech samples by two randomly selected algorithms - and instructed to choose either a preferred sample (score 1) or "can't decide" (score 0.5). A total of 30 subjects participated in the listening test, and the results are presented in Table 4 and in Table 7.

Table 4: Preference scores of DCUnet compared to other baseline models. The scores were obtained by calculating the relative frequency the subjects prefer one method to the other method. Hard/Medium/Easy denote 2.5/7.5/17.5 SNR conditions in dB, respectively. All statistics had significance of p<0.001. Complete pairwise preference scores are presented in Appendix H.

|  | Hard | Medium | Easy | Total |
| --- | --- | --- | --- | --- |
| DCUnet > Deep Feature Loss | 0.90 | 0.82 | 0.69 | 0.82 |
| DCUnet > Wavenet | 0.98 | 0.95 | 0.75 | 0.93 |
| DCUnet > SEGAN | 0.99 | 0.93 | 0.71 | 0.92 |

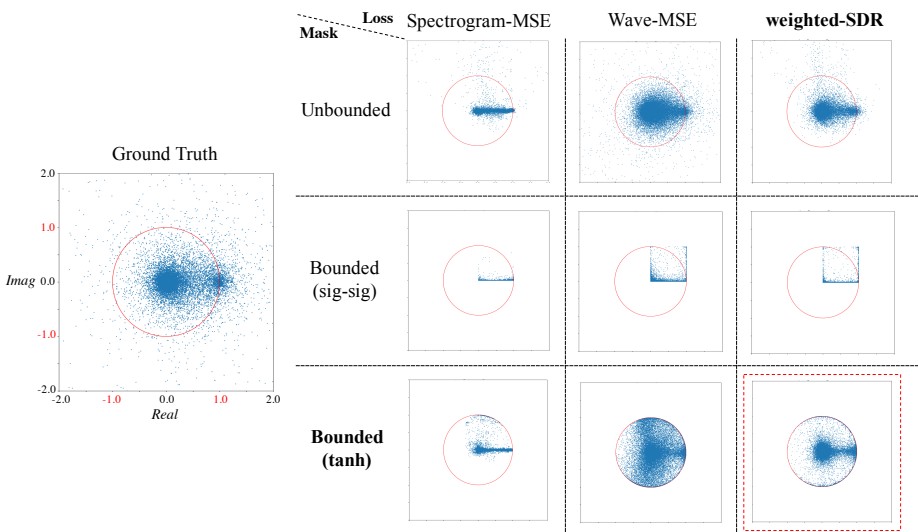

Figure 4: Scatter plots of estimated cRMs with 9 different mask and loss function configurations for a randomly picked noisy speech signal. Each scatter plot shows the distribution of complex values from an estimated cRM. The leftmost plot is from the cIRM for the given input. We can observe that most real-values are distributed around 0 and 1, while being relatively sparse in between. The configuration that fits the most to this distribution pattern is observed in the red dotted box which is achieved by the combination of our proposed methods (Bounded (tanh) and weighted-SDR).

Table 4 shows that DCUnet clearly outperforms the other methods in terms of preference scores in every SNR condition. These differences are statistically significant as confirmed by pairwise one-tailed t-tests. Furthermore, the difference becomes more obvious as the input SNR condition gets worse, which supports our motivation that accurate phase estimation is even more important under harsh noisy conditions. This is further confirmed by in-depth quantitative analysis of the phase distance as described in Section 5 and Table 5.

## 5 IN-DEPTH ANALYSIS ON PHASE

In this section, we aim to provide constructive insights on phase estimation by analyzing how and why our proposed method is effective. We first visualized estimated complex masks with scatter plots in Figure 4 for each masking method and loss function configuration from Table 2. The plots clearly show that the Bounded (sig-sig) mask by Ephrat et al. fails to reflect the distribution of the complex ideal ratio mask (cIRM), whereas the proposed Bounded (tanh) or Unbounded mask tries to follow the ground truth. One interesting finding is that, while time-domain loss functions such as weighted-SDR or Wave-MSE can capture the distribution of the ground truth, Spectrogram-MSE fails to capture the distribution of the cIRM. More specifically, when we use Spectrogram-MSE, the imaginary values of the predicted cRMs are almost always zero, indicating that it ends up only

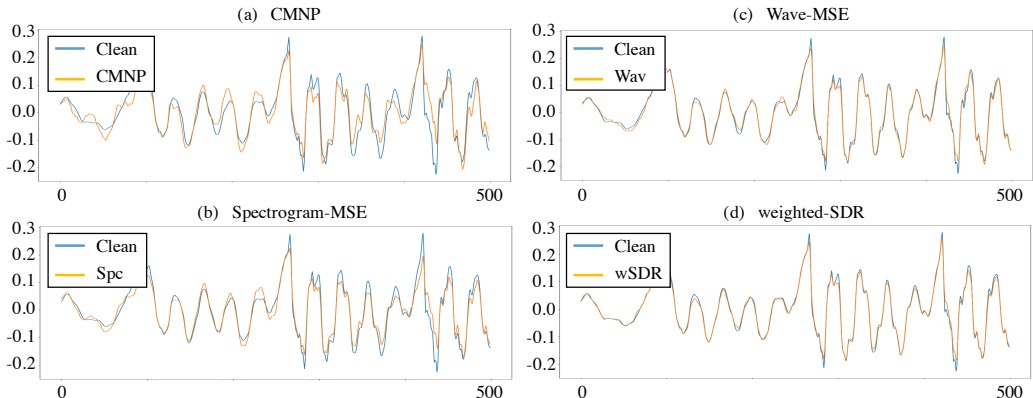

Figure 5: Illustration of four wave-plot segments of estimated speech with reference clean speech. (a) shows the wave-plot of synthesized speech with clean magnitude speech and noisy phase (CMNP). Three other cases show the wave-plots of estimated speech with different loss functions (b) Spectrogram-MSE, (c) Wave-MSE, and (d) weighted-SDR.

scaling the magnitude of noisy speech and fails to correct the phase of noisy speech with rotations (e.g., $(X_{t,f}^{real} + iX_{t,f}^{imag})(a + i \cdot 0) = a(X_{t,f}^{real} + iX_{t,f}^{imag}))$.

In order to demonstrate this effect in an alternate perspective, we also plotted estimated waveforms for each loss function in Figure 5. As one can notice from Figure 5 (c) & (d), estimated speech from models optimized with time-domain loss functions are well-aligned with clean speech. However, Figure 5 (a) & (b) align poorly to the ground truth while showing almost identical estimations. Note that Figure 5 (a) is the best possible result without estimating the phase of clean speech. This again confirms that optimizing the model with Spectrogram-MSE makes it difficult to learn phase correction, meaning that the model ends up reusing noisy phase just like conventional approaches.

To explicitly support these observations, we would need a quantitative measure for phase estimation. Here, we define the phase distance between target spectrogram ($A$) and estimated spectrogram ($B$) as the weighted average of angle between corresponding complex TF bins, where each bin is weighted by the magnitude of target speech ($|A_{t,f}|$) to emphasize the relative importance of each TF bin. Phase distance is formulated as the following:

$$PhaseDist(A, B) = \sum_{t,f} \frac{|A_{t,f}|}{\sum_{t',f'} |A_{t',f'}|} \angle(A_{t,f}, B_{t,f}) \tag{8}$$

where, $\angle(A_{t,f}, B_{t,f})$ represents the angle between $A_{t,f}$ and $B_{t,f}$, having a range of [0,180].

The phase distance between clean and noisy speech (**PhaseDist(C, N)**) and the phase distance between clean and estimated speech (**PhaseDist(C, E)**) are presented in Table 5. The results show that the best phase improvement (Phase Improvement = PhaseDist(C, N) − PhaseDist(C, E)) is obtained with wSDR loss under every SNR condition. Also Spc loss gives the worst results, again reinforcing our observation. Analysis between the phase improvement and performance improvement is further discussed in Appendix G.

## 6 CONCLUSION

In this paper, we proposed Deep Complex U-Net which combines two models to deal with complex-valued spectrograms for speech enhancement. In doing so, we designed a new complex-valued masking method optimized with a novel loss function, weighted-SDR loss. Through ablation studies, we showed that the proposed approaches are effective for more precise phase estimation, resulting in state-of-the-art performance for speech enhancement. Furthermore, we conducted both quantitative and qualitative studies and demonstrated that the proposed method is consistently superior to the previously proposed algorithms.

Table 5: Phase distance and phase improvement under four different SNR conditions. The Spc, Wav, and wSDR column show results from models trained with three different objectives Spectrogram-MSE, Wave-MSE, and weighted-SDR, respectively.

| SNR (dB) | PhaseDist(C, N) | PhaseDist(C, E) | | | Phase Improvement | | |
|---|---|---|---|---|---|---|---|
| | | Spc | Wav | wSDR | Spc | Wav | wSDR |
| 2.5 | 14.521° | 10.512° | 9.288° | **7.807°** | 4.009° | 5.233° | **6.714°** |
| 7.5 | 10.066° | 7.548° | 6.919° | **5.658°** | 2.518° | 3.147° | **4.408°** |
| 12.5 | 7.197° | 5.455° | 5.105° | **4.215°** | 1.742° | 2.092° | **2.982°** |
| 17.5 | 4.853° | 3.949° | 3.872° | **3.151°** | 0.905° | 0.981° | **1.702°** |

In the near future, we plan to apply our system to various separation tasks such as speaker separation or music source separation. Another important direction is to extend the proposed model to deal with multichannel audio since accurate estimation of phase is even more critical in multichannel environments (Wang et al., 2018). Apart from separation, our approach can be generalized to various audio-related tasks such as dereverberation, bandwidth extension or phase estimation networks for text-to-speech systems. Taking advantage of sequence modeling, it may also be interesting to find further extensions with complex-valued LSTMs (Arjovsky et al., 2016; Wisdom et al., 2016).

ACKNOWLEDGEMENTS

This work was partly supported by Next-Generation Information Computing Development Program through the National Research Foundation of Korea (NRF) funded by the Ministry of Science and ICT (MSIT; Grant No. NRF-2017M3C4A7078548), partly supported by National Research Foundation of Korea (NRF) funded by the Korea government (MSIT; Grant No. NRF-2017R1E1A1A01076284), and partly supported by the Creative Industrial Technology Development Program (10053249) funded by the Ministry of Trade, Industry and Energy (MOTIE, Korea).

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

## APPENDIX

### A    REAL-VALUED CONVOLUTION & COMPLEX-VALUED CONVOLUTION

In this section, we address the difference between the real-valued convolution and the complex-valued convolution. Given a complex-valued convolution filter $W = A + iB$ with real-valued matrices $A$ and $B$, the complex-valued convolution can be interpreted as two different real-valued convolution operations with shared parameters, as illustrated in Figure 6 (b). For a fixed number of $\#Channel\ product = \#Input\ channel(M) \times \#Output\ channel(N)$, the number of parameters of the complex-valued convolution becomes double of that of a real-valued convolution. Considering this fact, we built the pair of a real-valued network and a complex-valued network with the same number of parameters by reducing $\#Channel\ product$ of complex-valued convolution by half for a fair comparison. The detail of models reflecting this configuration is explained in Appendix B.

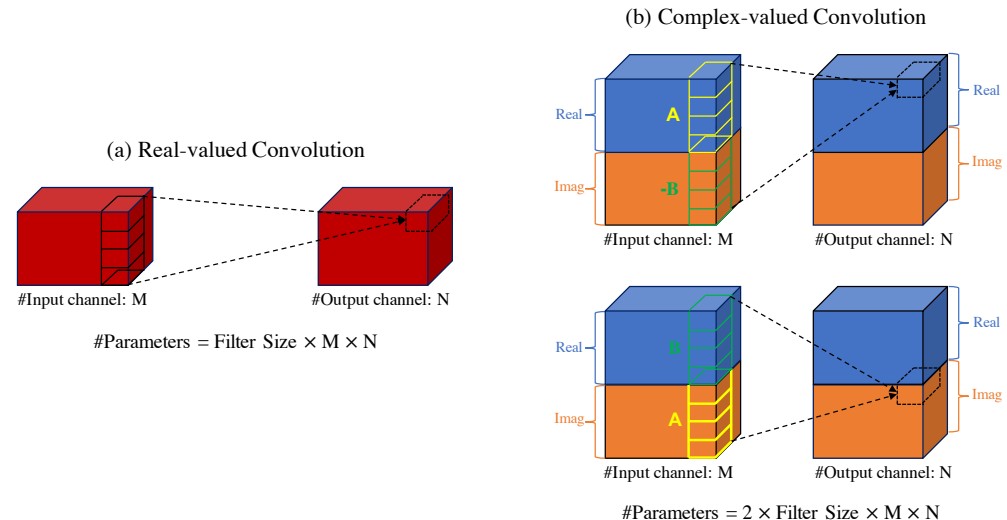

Figure 6: Illustration of (a) real-valued convolution and (b) complex-valued convolution.

### B    MODEL ARCHITECTURE

In this section, we describe three different model architectures (DCUnet-20 (#params: 3.5M), DCUnet-16 (#params: 2.3M), and DCUnet-10 (#params: 1.4M)) each in complex-valued network setting and real-valued network setting in Figure 7, 8, 9. Both complex-valued network ($\mathbb{C}$) and real-valued network ($\mathbb{R}$) have the same size of convolution filters with different number of channels to set the parameter equally. The largest model, Large-DCUnet-20, in Table 1 is also described in Figure 10. Every convolution operation is followed by batch normalization and an activation function as described in Figure 11. For the complex-valued network, the complex-valued version of batch normalization and activation function was used following Deep Complex Networks (Trabelsi et al., 2018). Note that in the very last layer of every model the batch normalization and leaky ReLU activation was not used and non-linearity function for mask was applied instead. The real-valued network configuration was not considered in the case of largest model.

### C    PROPERTIES OF WEIGHTED-SDR LOSS

In this section, we summarize the properties of the proposed weighted-SDR loss. First, we show that the range of weighted-SDR loss is bounded and explain the conditions under which the minimum value is obtained. Next, we explain the gradients in the case of noise-only input.

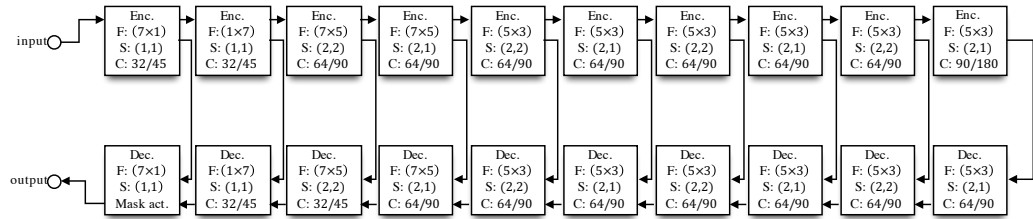

Figure 7: DCUnet-20 (20-layer): a model with 20 convolutional layers.

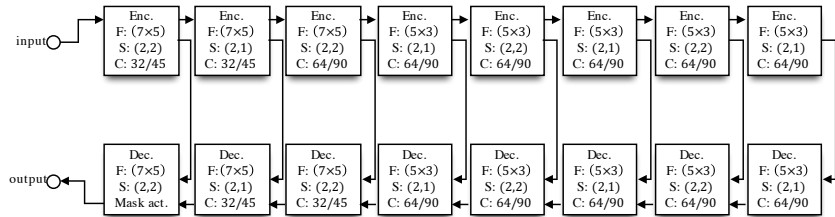

Figure 8: DCUnet-16 (16-layer): a model with 16 convolutional layers.

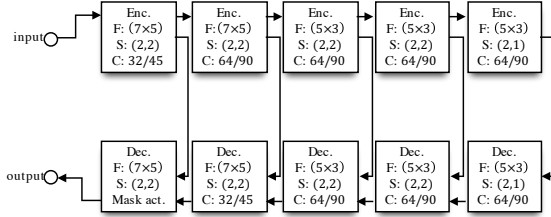

Figure 9: DCUnet-10 (10-layer): a model with 10 convolutional layers.

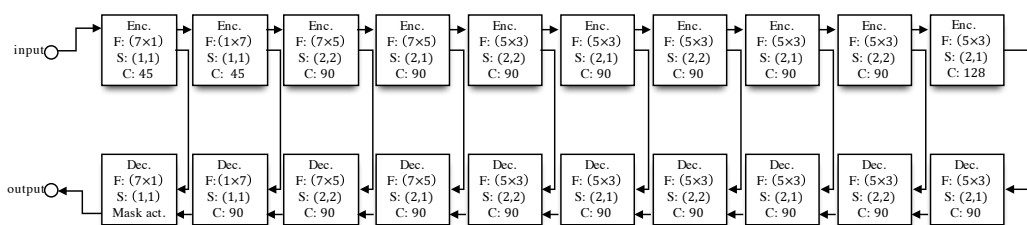

Figure 10: Large-DCUnet-20: a model with 20 convolutional layers with more number of channels per layer.

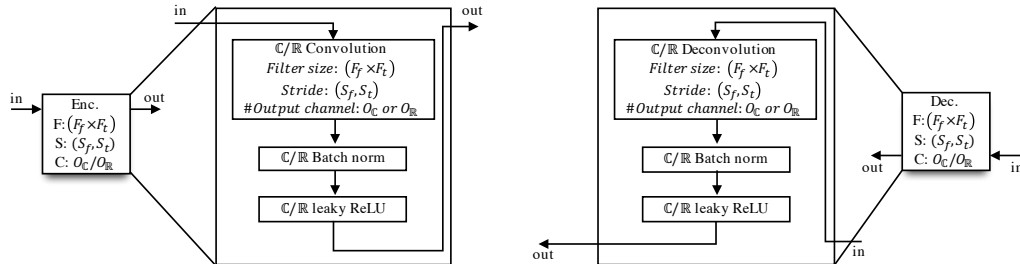

Figure 11: Description of encoder and decoder block. $F_f$ and $F_t$ denote the convolution filter size along the frequency and time axis, respectively. $S_f$ and $S_t$ denote the stride size of convolution filter along the frequency and time axis, respectively. $O_{\mathbb{C}}$ and $O_{\mathbb{R}}$ denote the different number of channels in complex-valued network setting and real-valued network setting, respectively. The number of channels of $O_{\mathbb{R}}$ is set to be roughly $\sqrt{2}$ times the number of channels of $O_{\mathbb{C}}$ so that the number of trainable parameters of real-valued network and complex-valued network becomes approximately the same.

Let $x$ denotes noisy speech with $T$ time step, $y$ denotes target source and $\hat{y}$ denotes estimated source. Then, $loss_{wSDR}(x, y, \hat{y})$ is defined as follows:

$$loss_{wSDR}(x, y, \hat{y}) = -\alpha \frac{<y, \hat{y}>}{\|y\|\|\hat{y}\|} - (1 - \alpha)\frac{<x - y, x - \hat{y}>}{\|x - y\|\|x - \hat{y}\|} \quad (9)$$

where, $\alpha$ is the energy ratio between target source and noise, i.e., $\|y\|^2 / (\|y\|^2 + \|x - y\|^2)$.

**Proposition 1.** $loss_{wSDR}(x, y, \hat{y})$ is bounded on [-1,1] . Moreover, for fixed $y \neq 0$ and $x - y \neq 0$, the minimum value -1 can only be attained when $\hat{y} = y$, if $x \neq cy$ for $\forall c \in \mathbb{R}$.

*Proof.* Cauchy-Schwarz inequality states that for $a \in \mathbb{R}^T$ and $b \in \mathbb{R}^T$, $-\|a\|\|b\| \leq <a, b> \leq \|a\|\|b\|$. By this inequality, [-1,1] becomes the range of $loss_{wSDR}$. To attain the minimum value, the equality condition of the Cauchy-Schwarz inequality must be satisfied. This equality condition is equivalent to $b = 0$ or $a = tb$, for $\exists t \in \mathbb{R}$. Applying the equality condition with the assumption $(y \neq 0, x - y \neq 0)$ to Equation 9 leads to $\hat{y} = t_1 y$ and $x - \hat{y} = t_2(x - y)$, $for \exists t_1 \in \mathbb{R}$ and $\exists t_2 \in \mathbb{R}$. By adding these two equations, we can get $(1 - t_1)x = (t_1 - t_2)y$. By the assumption $x \neq cy$, which is generally satisfied for large $T$, we can conclude $t_1 = 1, t_1 = t_2$ must be satisfied when the minimum value is attained. $\square$

The following property of the weighted-SDR loss shows that the network can also learn from noise-only training data. In experiments, we add small number $\epsilon$ to denominators of Equation 9. Thus for the case of $y = 0$, Equation 9 becomes

$$loss_{wSDR}(x, 0, \hat{y}) = -\frac{<x, x - \hat{y}>}{\|x\|\|x - \hat{y}\| + \epsilon} \quad (10)$$

**Proposition 2.** When we parameterize $\hat{y} = g_\theta(x)$, the $loss_{wSDR}(x, y, g_\theta(x))$ has a non-zero gradient with respect to $\theta$ even if the target source $y$ is empty.

*Proof.* We can calculate partial derivatives as follows:

$$\frac{\partial loss_{wSDR}(x, y, g_\theta(x))}{\partial \theta}\bigg|_{(x=x_0, y=0, \theta=\theta_0)} = \frac{\partial loss_{wSDR}(x_0, 0, g_\theta(x_0))}{\partial \theta}\bigg|_{\theta=\theta_0}$$

$$= <\frac{x_0}{\|x_0\|}, \frac{\partial}{\partial \theta}\{\frac{x_0 - g_\theta(x_0)}{\|x_0 - g_\theta(x_0)\| + \epsilon/\|x_0\|}\}\bigg|_{\theta=\theta_0} > \quad (11)$$

Thus, the non-zero gradients with respect to $\theta$ can be back-propagated. $\square$

## D   IRREDUCIBLE ERRORS

In this section, we illustrate two possible irreducible errors. Figure 12 (a) shows the irreducible phase error due to lack of phase estimation. Figure 12 (b) shows the irreducible error induced when bounding the range of mask. Not bounding the range of the mask makes the problem well-posed but it may suffer from the wide range of optimization search space because of the lack of prior knowledge on the distribution of cIRM.

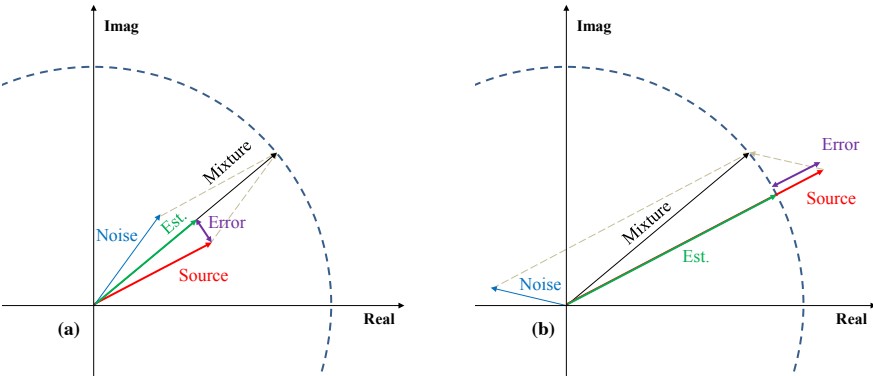

Figure 12: Two cases of possible irreducible error types in a TF bin in spectrogram. (a) The case where phase is not estimated and phase of mixture is reused. Even if the magnitude estimation is assumed to be perfect, there is still an irreducible phase error between the true source and the estimation. (b) The case where the magnitude of an estimated mask is bounded to 1. This again induces an irreducible error since there can be sources which have a higher magnitude than the magnitude of a mixture.

## E   SCATTER PLOTS OF CIRM

The scatter plots of cIRM from training set is shown in Figure 13. We show four different scatter plots according to their SNR values of mixture (0, 5, 10, and 15 (dB)). Each scattered point of cIRM, $M_{t,f}$, is defined as follows:

$$M_{t,f} = \frac{|Y_{t,f}|}{|X_{t,f}|} e^{i(\theta_{Y_{t,f}} - \theta_{X_{t,f}})} \tag{12}$$

The scattered points near origin indicate the TF bins where the value of $|Y_{t,f}|$ is significantly small compared to $|X_{t,f}|$. Therefore, those TF bins can be interpreted as the bins dominated with noise rather than source. On the other hand, the scattered points near $(1,0)$ indicates the TF bins where the value of $|Y_{t,f}|$ is almost the same as $|X_{t,f}|$. In this case, those TF bins can be interpreted as the bins dominated with source rather than noise. Therefore, as SNR becomes higher, the amount of TF bins dominated with clean source becomes larger compared to the lower SNR cases, and consequently the portion of real part close to 1 becomes larger as in Figure 13.

## F   VISUALIZATION OF ESTIMATED PHASE

In this section, we show a supplementary visualization of phase of estimated speech. Although the raw phase information itself does not show a distinctive pattern, the hidden structure can be revealed with group delay, which is the negative derivative of the phase along frequency axis (Yegnanarayana & Murthy, 1992). With this technique, the phase information can be explicitly shown as in Figure 14. Figure 14 (d) shows the group delay of clean speech and the corresponding magnitude is shown in Figure 14 (a). The two representations shows that the group delay of phase has a similar structure to that of magnitude spectrogram. The estimated phase by our model is shown in in Figure 14 (c). While the group delay of noisy speech (Figure 14 (b)) does not show a distinctive harmonic pattern, our estimation show the harmonic pattern similar to the group delay of clean speech, as shown in the yellow boxes in Figure 14 (c) and (d).

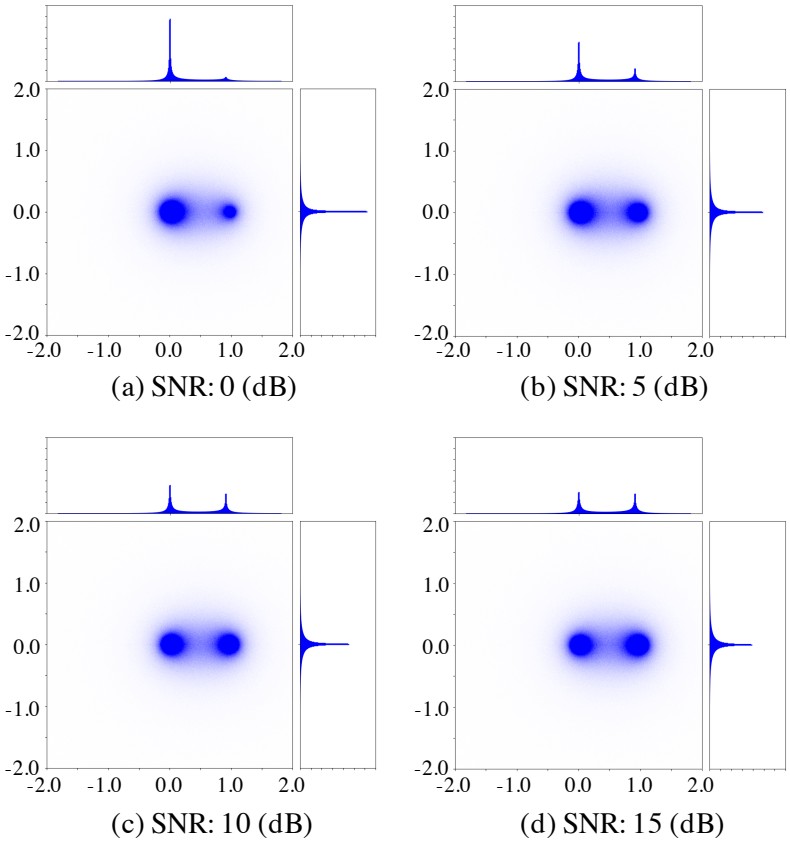

Figure 13: Four different scatter plots of cIRM according to the four different SNR values of input mixture in training set, (a) SNR: 0 (dB) (b) SNR: 5 (dB) (c) SNR: 10 (dB) (d) SNR: 15 (dB).

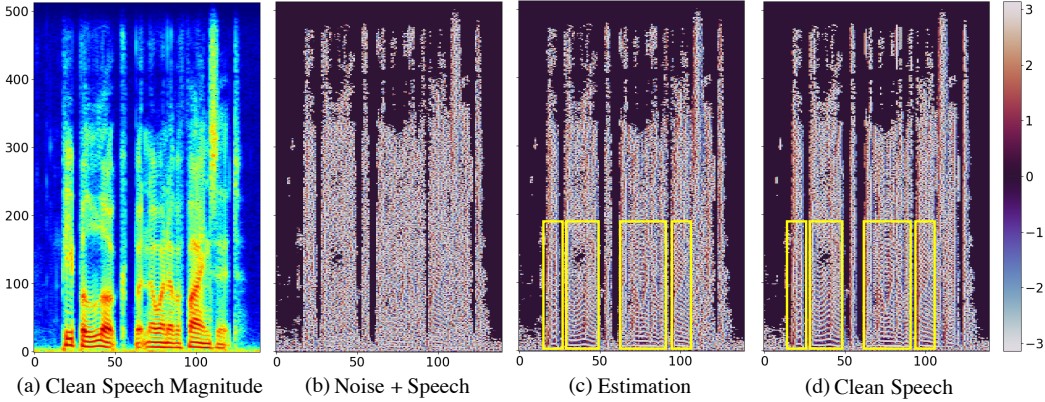

(a) Clean Speech Magnitude    (b) Noise + Speech    (c) Estimation    (d) Clean Speech

Figure 14: Illustration of phase group delay. The group delay of the estimated phase from our model (c) shows the similar pattern to that of clean speech (d). For better visualization, we only show the TF bins where the magnitude of clean speech spectrogram exceeds certain threshold 0.01.

## G   IMPORTANCE OF PHASE ESTIMATION

In this section, to show the limitation of conventional approach (without phase estimation), we emphasize that the phase estimation is important, especially under low SNR condition (harsh condition).

We first make an assumption that the estimation of phase information becomes more important when the given mixture has low SNR. Our reasoning behind this assumption is that if the SNR of a given mixture is low, the irreducible phase error is likely to be greater, hence a more room for improvement with phase estimation as illustrated in Figure 15. This can also be verified in Table 5 columns **PhaseDist(C, N)** and **Phase Improvement** where the values of both columns increase as SNR becomes higher.

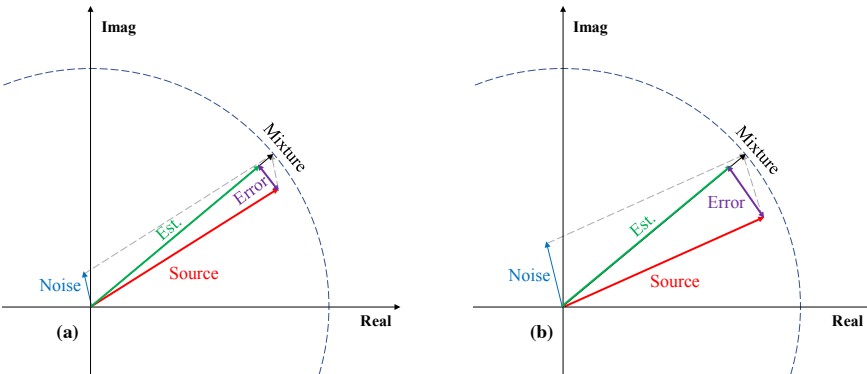

Figure 15: (a) The case where SNR of given mixture is high. In this case the source is likely to be dominant in the mixture. Therefore it is relatively easier to estimate ground truth source with better precision even when the phase is not estimated. (b) The case where SNR of given mixture is low. In this case the source is not dominant in the mixture. Therefore, the irreducible phase error is likely to be higher in low SNR conditions than higher SNR conditions. Under this circumstance, we assume the lack of phase estimation will result in a particularly bad system performance.

To empirically show the importance of phase estimation, we show correlation between phase improvement and performance difference between the conventional method (without phase estimation) and our proposed method (with phase estimation) in Table 6. The performance difference was calculated by simply subtracting the evaluation results of conventional method from the evaluation results of our method with phase estimation. For fair comparison, both conventional method (RM$\mathbb{R}$n) and proposed method (cRM$\mathbb{C}$n) were set to have the same number of parameters. Also, both models were trained with weighted-SDR loss. The results show that when the SNR is low, both the phase improvement and the performance difference are relatively higher than the results from higher SNR conditions. Furthermore, almost all results show an incremental increase of phase improvement and performance difference as the SNR decreases, which agrees on our assumption. Therefore we believe that phase estimation is important especially in harsh noisy conditions (low SNR conditions).

Table 6: The performance difference between conventional method (without phase estimation) and our method (with phase estimation). The performance difference is presented with four different SNR values of mixture in test set.

| SNR (dB) | Phase Improvement | Performance Difference | | | | |
|---|---|---|---|---|---|---|
| | | PESQ | CSIG | CBAK | COVL | SSNR |
| 2.5 | 6.714° | 0.06 | 0.44 | 0.14 | 0.21 | 5.32 |
| 7.5 | 4.408° | 0.03 | 0.42 | 0.11 | 0.19 | 5.21 |
| 12.5 | 2.982° | 0.05 | 0.39 | 0.11 | 0.18 | 4.96 |
| 17.5 | 1.702° | 0.05 | 0.30 | 0.09 | 0.13 | 3.93 |

## H COMPLETE QUALITATIVE RESULTS

Table 7: Pairwise preference scores of four models including DCUnet. The scores are obtained by calculating the relative frequency the subjects prefer one method to the other method. Hard/Medium/Easy denote 2.5/7.5/17.5 SNR conditions in dB, respectively. Significance for each statistic is also described (n.s.: not significant, $*$: p<0.05, $**$: p<0.01, $***$: p<0.001).

| | Hard | Medium | Easy | Total |
|---|---|---|---|---|
| DCUnet > Deep Feature Loss | 0.90 (***) | 0.82 (***) | 0.69 (***) | 0.82 (***) |
| DCUnet > Wavenet | 0.98 (***) | 0.95 (***) | 0.75 (***) | 0.93 (***) |
| DCUnet > SEGAN | 0.99 (***) | 0.93 (***) | 0.71 (***) | 0.92 (***) |
| Deep Feature Loss > Wavenet | 0.67 (***) | 0.83 (***) | 0.48 (n.s.) | 0.74 (***) |
| Deep Feature Loss > SEGAN | 0.72 (***) | 0.81 (***) | 0.40 (**) | 0.73 (***) |
| SEGAN > Wavenet | 0.53 (n.s.) | 0.58 (**) | 0.50 (n.s.) | 0.56 (**) |

