# OpenReview forum: "Phase-Aware Speech Enhancement with Deep Complex U-Net"
_ICLR.cc/2019/Conference_

### Official Review · AnonReviewer3 · 2018-11-03
**generally good paper on speech enhancement using complex operations**

**Rating:** 7
**Confidence:** 4

**Review:**

This paper used a complex-valued network to learn the modified complex ratio mask with a weighted SDR loss for the speech enhancement task. It can get good enhancement performance.

For me, the complex-valued network is already there and weighted SDR loss is not difficult to think. The modified complex ratio mask is a bit interesting. However, I think it better to compare with [Donald S Williamson et al] where the hyperbolic tangent compression is used.

Apart from the objective metrics, a human listening test using MOS or preference score should be conducted.

On Fig 3, the unbounded complex mask might suffer from the infinity problem leading to training failure. However, on table 2, the performance of the unbounded mask is quite close to your method. It is a bit strange for me.

The total idea is good, but the novelty is not much.

---

> ### Author Response · Authors · 2018-11-10
> **Response to Reviewer#3**
>
> Thank you for your review and comments.
>
> Below are the responses to each of your comments.
>
> For me, the complex-valued network is already there and weighted SDR loss is not difficult to think. The modified complex ratio mask is a bit interesting. However, I think it better to compare with [Donald S Williamson et al] where the hyperbolic tangent compression is used.
> -> Answer:
> Thank you for your suggestion. In fact, we tried this before. Since the perceptual quality of hyperbolic tangent compression was not good compared to the other masking methods, we did not add the results in our manuscript. However, as you suggested, we think it is fair to have the actual quantitative results to compare, and thus we are currently retraining the network using hyperbolic tangent compression and will report the result as soon as the training finishes.
>
>
> Apart from the objective metrics, a human listening test using MOS or preference score should be conducted.
> -> Answer:
> Thank you for your suggestion. We will conduct a user listening study with random samples from the test dataset and update the manuscript by adding the results as soon as possible.
>
>
> On Fig 3, the unbounded complex mask might suffer from the infinity problem leading to training failure. However, on table 2, the performance of the unbounded mask is quite close to your method. It is a bit strange for me.
> -> Answer:
> Although it may seem to theoretically suffer from the infinite search space, the real distribution of the ideal complex masks are most likely bounded to a small finite region, which is likely to help alleviate the problem. In our case, we had no such problems when training our models.

---

> > ### Author Response · Authors · 2018-11-23
> > **Response to Reviewer 3 regarding the mask performance.**
> >
> > Reflecting the concern by the Reviewer 3, we conducted further experiments by implementing the ‘tanh compression’ mask.
> > As expected, our method gave better performance by every quantitative measure.
> >
> >                                     CSIG      CBAK    COVL    PESQ    SSNR
> > BDT (ours)                 4.18        3.77      3.63       3.06     13.29
> > TanhCompression    4.11       3.33      3.56       3.01       7.01

---

### Official Review · AnonReviewer2 · 2018-11-04
**review of "Phase-Aware Speech Enhancement with Deep Complex U-Net"**

**Rating:** 7
**Confidence:** 4

**Review:**

This paper tackles one of important speech enhancement issues of how to predict phase information. The authors work on this problem based on three novel techniques, one is to use complex U-net, second is to propose a new complex mask representation, which is well bounded and well model complex mask distribution, and the last is an objective function motivated by SDR. The paper is well written, and also shows the experimental effectiveness of the proposed method by analyzing these three novel techniques and also by comparing the method with other speech enhancement methods. My major concern about this paper is that this paper is a little bit too specific to the speech enhancement applications, which will not be accepted with so many researches in the major ICLR community. My suggestion is to describe some potential applications of this method to the other (speech) applications including speech separation, noise-robust front-end for ASR, TTS, or other speech analysis, and also discuss the possibility of extending this method for multichannel input. I’m more interested in the multichannel enhancement because the phase (difference) is critical in this scenario.

Comments:
- Introduction: It’s better to cite and discuss the paper of “E. Hakan et al, “Phase-sensitive and recognition-boosted speech separation using deep recurrent neural networks,” Proc. ICASSP’15, pp. 708--712 (2015). This paper is one of the first studies tries to incorporate the phase information to DNN based speech enhancement.
- Several researchers prefer to use LSTM based enhancement method. Please discuss wether this method (objective function and complex masks) can be applied to complex extensions of LSTMs instead of complex U-net.
- Page 2, the first paragraph: You may also refer https://arxiv.org/abs/1810.01395
- Page 3, it’s better to explicitly mention that h = x + i y
- Section 3.3: discuss how we treat STFT/iSTFT operations under a computational graph representation. It is not so obvious.
- Section 3.3: again it’s better to mention E. Hakan’s method here.
- Page 6 footnote: I cannot access to the URL. Please check it.
- Experiments: I think it would be more interesting to add SDR (using speech and noise as a source) to the experimental measure. Some people use SDR as a speech enhancement measure, and I’m expecting that this method can have more reasonable performance since it is optimized based on wSDR.

---

> ### Author Response · Authors · 2018-11-10
> **Response to Reviewer#2**
>
> We thank the reviewer for the extensive comments, which were very constructive and helpful for building a better paper.
>
> Below are the responses to each of your comments.
>
> My major concern about this paper is that this paper is a little bit too specific to the speech enhancement applications, which will not be accepted with so many researches in the major ICLR community. My suggestion is to describe some potential applications of this method to the other (speech) applications including speech separation, noise-robust front-end for ASR, TTS, or other speech analysis, and also discuss the possibility of extending this method for multichannel input.
> -> Answer:
> Thank you for your suggestion. We will add more explanations on potential applications and describe speech enhancement as a fundamental problem for general audio tasks.
>
>
> I’m more interested in the multichannel enhancement because the phase (difference) is critical in this scenario.
> -> Answer:
> We acknowledge the importance of such scenarios and also are interested in studying the case. We will add a discussion in the future work.
>
>
> - Introduction: It’s better to cite and discuss the paper of “E. Hakan et al, “Phase-sensitive and recognition-boosted speech separation using deep recurrent neural networks,” Proc. ICASSP’15, pp. 708--712 (2015). This paper is one of the first studies tries to incorporate the phase information to DNN based speech enhancement.
> -> Answer:
> Thank you for the suggestion. We will add sentences to the Introduction as one of the initial dnn-based approaches incorporating phase information.
>
>
> - Several researchers prefer to use LSTM based enhancement method. Please discuss whether this method (objective function and complex masks) can be applied to complex extensions of LSTMs instead of complex U-net.
> -> Answer:
> Thank you for your idea for extension. As you mentioned, our objective function and complex masking method can be applied to complex-valued LSTMs, which will be expected to be effective for sequential representation learning and potentially improve the performance. We will add this discussion to the Conclusion.
>
>
> - Page 2, the first paragraph: You may also refer https://arxiv.org/abs/1810.01395
> -> Answer:
> Thank you for notifying us, but we were not aware of this paper since the deadline was the end of September. As it is very relevant in terms of phase estimation, we will refer to this work in the paper.
>
>
> - Page 3, it’s better to explicitly mention that h = x + i y
> -> Answer:
> We will fix this in the updated version soon.
>
>
> - Section 3.3: discuss how we treat STFT/iSTFT operations under a computational graph representation. It is not so obvious.
> -> Answer:
> We will add the description in Section 3.3.
>
>
> - Section 3.3: again it’s better to mention E. Hakan’s method here.
> -> Answer:
> We will refer to the E. Haken’s method in Section 3.2 as it is more related to the masking method.
>
>
> - Page 6 footnote: I cannot access to the URL. Please check it.
> -> Answer:
> We think the URL doesn’t work because the underbars have been removed from from the hyperlink. We will fix this.
>
> - Experiments: I think it would be more interesting to add SDR (using speech and noise as a source) to the experimental measure. Some people use SDR as a speech enhancement measure, and I’m expecting that this method can have more reasonable performance since it is optimized based on wSDR.
> -> Answer:
> For quick comparison, we evaluated SDR for DCUnet-20 with the BDT mask setting, and obtained the following results:
> 	       Spc      Wav     wSDR
> SDR      23.17 | 23.99 | 24.16
> SSNR     9.54  | 12.34 | 13.29
> As expected, wSDR yielded the best performance among the three loss terms compared in the paper.
> However, we would like to note that SDR is essentially scale-invariant SNR, and since wSDR loss tries to fit the scale of the target source, it leads to maximizing SNR (not scale-invariant) more than maximizing SDR. This can be confirmed by the fact that more dramatic improvement is observed in terms of SSNR rather than SDR.

---

### Official Review · AnonReviewer1 · 2018-11-05
**well written & rather experimental paper -- for the experts mostly**

**Rating:** 6
**Confidence:** 4

**Review:**

The paper is written, provides good description of the state-of-the-art and comprehensive experimental results.
The methological contribution is mild, essentially changing a buiding block in a state-of-the-art neural architecture.
The paper is for the expert audience mostly and is difficult to grasp without a good background on deep learning for speech enhancement.

---

> ### Author Response · Authors · 2018-11-10
> **Response to Reviewer#1**
>
> Thank you for your review.
>
> Below are the responses to each of your concerns.
>
> The methodological contribution is mild, essentially changing a building block in a state-of-the-art neural architecture.
> -> Answer:
> We agree that the change of building block (DCUNet) itself might be considered as a mild contribution with respect to methodological points. However, in addition to the modification, our main contributions include a novel masking approach and an advanced loss function design. We believe it is not trivial to successfully incorporate these components, as this results in remarkable performance improvement from the previously proposed methods. Furthermore, as far as we are concerned, this is the first work that enables efficient phase estimation using continuous regression with a complex-valued method.
>
>
> The paper is for the expert audience mostly and is difficult to grasp without a good background on deep learning for speech enhancement.
> -> Answer:
> Thank you for pointing this out. As most of the general audience may have less understanding about speech signal modeling, we added additional explanations to the Introduction we consider fundamental for a wider range of audience.

---

### Author Response · Authors · 2018-11-23
**Our revised paper has been uploaded.**

We would like to thank all the reviewers for their fruitful comments and suggestions that help make our paper more complete and comprehensive.
We have uploaded a newly revised paper reflecting almost all the comments, concerns and suggestions.
We mainly focused on revising the Introduction and Conclusion sections to make the manuscript more comprehensible to general audiences by clarifying the motivation of our work and by describing the potential applications of our work.
In addition, we conducted subjective listening tests and demonstrated that the proposed approach yields superior performance qualitatively.
If there are any further recommendations for the revised paper, we would like to reflect those until the due date.

---

### Author Response · Authors · 2019-04-28
**Retracted from ICLR2019.**

Dear readers, this is the announcement of the retraction of our paper "Phase-Aware Speech Enhancement with Deep Complex U-Net" from ICLR2019.

First, we thank you for the interest in our work.

We are truly sorry, however, to inform that a significant error was found by ourselves in the experimental process of our paper “Phase-aware Speech Enhancement with Deep Complex U-Net”, which was accepted for ICLR2019. After careful examinations, we therefore have made a decision to retract the accepted paper.

To be more specific about the error, we found out that the training data path was accidentally set to the evaluation data path which means the reported numbers in the table are utterly wrong (possibly overfitted results).

Lastly, we sincerely apologize for the mistake we made and promise it will not happen again.

Best regards,
Authors

---

> ### Public Comment · (anonymous) · 2019-07-17
> **Thank you for your honesty**
>
> Dear Authors,
>
> As a random reader of your paper, I really appreciate your honesty. Also a thank to the ICLR organizer for hosting papers in OpenReview. It is a great platform for a continual review!
>
> Thank you

---

### Meta-Review · Area_Chair1 · 2018-12-14
**Application specific paper, but well written with interesting evaluations and analysis**

**Confidence:** 5
**Recommendation:** Accept (Poster)

**Metareview:**

The authors propose an algorithm for enhancing noisy speech by also accounting for the phase information. This is done by adapting UNets to handle features defined in the complex space, and by adapting the loss function to improve an appropriate evaluation metric.

Strengths
- Modifies existing techniques well to better suit the domain for which the algorithm is being proposed. Modifications like extending UNet to complex Unet to deal with phase, redefining the mask and loss are all interesting improvements.
- Extensive results and analysis.

Weaknesses
- The work is centered around speech enhancement, and hence has limited focus.

Even though the paper is limited to speech enhancement, the reviewers agreed that the contributions made by the paper are significant and can help improve related applications like ASR. The paper is well written with interesting results and analysis. Therefore, it is recommended that the paper be accepted.